# Bone Marrow Adipocytes: A Link between Obesity and Bone Cancer

**DOI:** 10.3390/cancers13030364

**Published:** 2021-01-20

**Authors:** Michaela R. Reagan, Heather Fairfield, Clifford J. Rosen

**Affiliations:** 1Center for Molecular Medicine, Maine Medical Center Research Institute, Scarborough, Maine, ME 04074, USA; HFCampbell@mmc.org (H.F.); rosenc@mmc.org (C.J.R.); 2School of Medicine, Tufts University, Boston, MA 02111, USA; 3Graduate School of Biomedical Science and Engineering, University of Maine, Orono, ME 04469, USA

**Keywords:** myeloma, bone marrow adipocyte, breast cancer, prostate cancer, leukemia, adipocyte, BMAT, fatty acids, adipokine

## Abstract

**Simple Summary:**

This review discusses the important newly-established roles for bone marrow adipose tissue in cancer progression and highlights the research demonstrating great promise for clinically targeting the cells in oncology. Bone marrow adipose tissue expands during aging and in obesity. It primarily comprises bone marrow adipocytes (also known as fat cells) and can also contain other cells, such as pre-adipocytes, fibroblasts, macrophages, other immune cells, and endothelial cells. Bone marrow adipocytes are scattered throughout the hematopoietic or “red” marrow, or are densely packed in the marrow cavity, creating “yellow” marrow. Bone marrow biologists are interrogating many questions to understand the nature of bone marrow adipocytes, including how aging and obesity affect these cells; their origins, functions, and endocrine roles; and whether they can be targeted to treat osteoporosis. In parallel, and often in concert, cancer researchers are delineating the role of bone marrow adipocytes in oncology and their potential translational significance for future therapeutics.

**Abstract:**

Cancers that grow in the bone marrow are for most patients scary, painful, and incurable. These cancers are especially hard to treat due to the supportive microenvironment provided by the bone marrow niche in which they reside. New therapies designed to target tumor cells have extended the life expectancy for these patients, but better therapies are needed and new ideas for how to target these cancers are crucial. This need has led researchers to interrogate whether bone marrow adipocytes (BMAds), which increase in number and size during aging and in obesity, contribute to cancer initiation or progression within the bone marrow. Across the globe, the consensus in the field is a unified “yes”. However, how to target these adipocytes or the factors they produce and how BMAds interact with different tumor cells are open research questions. Herein, we review this research field, with the goal of accelerating research in the network of laboratories working in this area and attracting bright scientists with new perspectives and ideas to the field in order to bring about better therapies for patients with bone cancers.

## 1. Introduction

Bone cancer is one of the most destructive and painful manifestations of malignancy, and is often terminal for patients with tumors that originate in other locations and disseminate through the bloodstream to the bone marrow. Bone cancers can also result from a primary tumor within the bone, such as an osteosarcoma, or a hematopoietic malignancy that originates either within the bone marrow or elsewhere in the blood or lymph nodes, such as multiple myeloma, lymphoma, or leukemia. Bone cancers in general are reviewed throughout this Special Issue in depth, and thus this review will focus on the role of the bone marrow adipocyte (BMAd) in these malignancies. The BMAd derives from adipogenic differentiation of a mesenchymal stem cell (MSC) within the bone marrow, although as discussed below, the exact nature of the progenitor cell is still being investigated. The BMAd can be regulated by numerous factors, as previously reviewed [1,2]. Data from the Cawthorn group demonstrated that bone marrow adipose tissue (BMAT) is transcriptionally and functionally distinct from white, brown, and beige adipose tissues and that BMAT influences systemic glucose homeostasis [3]. Recent data from the Iwaniec and Turner team have demonstrated the long-term effects of rodent housing temperatures and the nervous system on BMAT. Housing mice at room temperature (22 °C) reduced their BMAT volumes as well as certain bone volumes compared to mice housed at thermoneutrality (32 °C) [4]. Interestingly, the sympathetic nervous system appeared to play a role in BMAT regulation, as treatment with the non-specific β-blocker propranolol (primarily β_1_ and β_2_ antagonists) prevented housing-temperature-associated differences in BMAT [5]. This effect did not appear to be an indirect effect resulting from improved bone parameters, since propranolol had only small effects on bone microarchitecture (increased trabecular number and decreased trabecular spacing) and did not attenuate premature bone loss induced by the colder room temperature housing [5]. The group found that the lower temperature housing reduced, whereas propranolol increased, expression of *Acacb*, the gene that codes for acetyl-CoA carboxylase, the rate-limiting step for fatty acid synthesis and a key regulator of β-oxidation [5].

Bone marrow adipocytes are also found to be distinct from other white, brown, or beige adipocytes, as can be seen in their distinctive responses to starvation and β3-agonist stimulation [6,7]. The biology of BMAds is further complicated by the fact that different bone marrow locations (e.g., vertebrae, femur, and tibia; distal versus proximal) have different types of BMAds, which respond differently to a variety of stimuli [1,8]. However, BMAds are also similar to white adipocytes in that they can store and release fatty acids and adipokines (such as adiponectin) and that they are predominantly unilocular [7]. Hundreds of genes and dozens of pathways have been identified as differentially expressed or activated in BMAT vs. white adipose tissue (WAT) based on microarray gene expression analysis of primary rabbit adipose tissues [7]. In their analysis, Craft et al. found that BMAT metabolism showed differences from WAT metabolism, including decreased glycerol content, insulin resistance, decreased fatty acid metabolism, and decreased thermoregulation. Gene expressions related to lipid synthesis and transport pathways (such as leptin, *FABP4*, *FABP5*, and interleukin 6 signal transducer (*IL6ST*)) were also reduced in BMAT although were still expressed, while cellular pathway analysis suggested reductions in fatty acid β-oxidation and oxidative phosphorylation in BMAT [7]. Other genes and proteins were found to be increased in BMAT compared to WAT, such as caveolin family proteins, collagens, and integrins, among many others. Ingenuity pathway analysis identified PPAR signaling as increased in BMAT [7,9]. Overall, this work demonstrates that the interactions between white adipocytes and tumor cells may be similar to interactions between BMAds and tumor cells, but that there are also likely crucial differences. The distinction between BMAds and other adipocytes is, thus, one important facet to consider when interpreting the roles of BMAds in cancer, since many of the reported in vitro data are based on white adipocytes rather than primary BMAds or MSC-derived BMAds. It would enhance the field to conduct experiments to compare tumor cell responses to a variety of types of adipocytes and to assess tumor growth, metastasis, migration, metabolic shifts, and drug responses in vitro to accurately clarify the effects of WAT and BMAT on cancer cells. In vivo models designed for this purpose would also be useful, although designing and conducting these studies would have significant challenges.

Few data are currently available comparing fresh, primary BMAds to those differentiated from bone marrow MSCs from any species. One reason for this is that primary BMAds are incredibly challenging to isolate and culture, although many laboratories, including ours, are working on optimizing this process now. The best option if primary BMAs are not available is to differentiate bone marrow MSCs into adipocytes in vitro, although these are still a simplified model of true primary BMAds. Moreover, considering the species, age, and sex of the BMAd or bone marrow donors, which may be limited based on the samples available, is important. Lastly, researchers can and do use primary white adipocyte cell lines, such as 3T3L1 cells, or other white adipocytes to gain insight into cell–cell interactions using in vitro co-cultures with tumor cells, and this may suffice as a first approximation of what occurs in human bodies. More effort should be put into understanding the differences between different types of adipocytes and their interactions with tumor cells by developing more realistic models, as well as better culture, isolation, and validation procedures and protocols that can be shared worldwide, as is being driven by the Bone Marrow Adipose Society (https://bma-society.org/) [10,11].

BMAd progenitor cells are leptin-receptor-positive (LepR+) MSCs, a population that also represents the main source of bone mesenchymal cells formed by adult bone marrow [12]. Studies in humans [13] and mice [14] suggest that obesity accelerates the differentiation of bone marrow MSCs by shifting their molecular phenotype and increasing insulin signaling while accelerating senescence in the marrow. Aging has also been shown to increase bone marrow adipogenesis in humans and rodents, and BMAT can sometimes correlate with bone volume loss or decreased bone compressive strength [15]. In mice, increased BMAT in aging was partially due to an increase in receptor activator of nuclear factor kappa B ligand (RANKL) and pre-adipocyte factor 1 (Pref1) co-positive pre-adipocytes during aging [16]. Fan et al. confirmed expression of RANKL in mature adipocytes, representing an important new avenue through which adipocytes can regulate osteoclasts and used a genetic deletion model of the PTH/PTHrP receptor (PTHR1) in MSCs with the Prx1-Cre mouse to demonstrate the ability for PTH signaling to modulate BMAT through effects on Pref1+RANKL+ marrow progenitors [17]. Almeida et al. demonstrated that aging increases adiposity in mice and that deletion of the peroxisome proliferator-activated receptor gamma (PPARγ) from early mesenchymal progenitors targeted by the Prx1-Cre transgene slowed aging-related marrow adipogenesis in 22-month-old female mice [18]. Interestingly, their work demonstrated that despite this decreased marrow adiposity, trabecular bone loss associated with aging still occurred as normal, suggesting that bone loss during aging is not completely dependent on increased BMAT or changes in the pre-adipocyte or pre-osteoblast MSC [18]. In contrast, they found that PPARγ expression in MSCs is important for cortical bone integrity and limiting cortical porosity during aging [18]. Most recently, a population of cells termed marrow adipogenic lineage precursors (MALPs), cells that are adiponectin-positive but without lipid droplets, have been defined using single-cell transcriptomics from the Ling Qin laboratory [19]. Overall, although we have learned a great deal about BMAT, BMAd progenitors, and the BMAT–bone relationship, many factors such as sex, hormones, sympathetic nerve signaling, temperature, and age contribute to this relationship, demonstrating that more research is needed to fully understand BMAT biology.

Nevertheless, BMAds and their progenitors are now recognized as essential mediators of the growth, progression, and metastasis of cancers within the bone marrow through many factors (Table 1). Given the lethality of these cancers, the need to quickly understand and target BMAds in cancer is pressing. Herein, we summarize the most recent and important findings regarding the field’s knowledge of BMAds in cancer within the bone microenvironment. Overall, independent of tumor type, the data suggest that BMAds support tumor cells through a myriad of factors, making these important potential targets for cancer therapy. As aging and obesity are risk factors for a number of cancers, and both are associated with increased BMAT, important questions in the field remain: how does aging biology and obesity pathophysiology affect BMAT, and how does aged or obese patient BMAT differentially affect the propensity for tumor cells to develop, proliferate, metastasize, evade the immune system, recruit blood vessels, or resist treatment? Moreover, as many cancer therapies and treatments (such as dexamethasone [20] and irradiation [21]) also increase BMAT, the importance of understanding how BMAT contributes to cancer progression cannot be ignored.

BMAds (bone marrow adipocytes); IL-6 (interleukin 6); IL-17 (interleukin 17); MCP1 (monocyte chemoattractant protein-1); SDF1 (stromal cell-derived factor 1); SCF (stem cell factor); IL-7 (interleukin 7); IL-15 (interleukin 15); IL-34 (interleukin 34); M-CSF (macrophage colony-stimulating factor); BMP-4 (Bone morphogenetic protein 4); CCL-19 (Chemokine ligand 19); CD36 (cluster of differentiation 36); FABP4 (fatty acid binding protein 4); FABP5 (fatty acid binding protein 5); CPT1 (Carnitine palmitoyltransferase I); HIF1a (hypoxia-inducible factor 1-alpha); ER (endoplasmic reticulum); JAK (janus kinase); STAT (signal transducer and activator of transcription protein); Atg3 (autophagy related 3); Atg5 (autophagy related 5); LC3 I-II (forms of the microtubule-associated proteins 1A/1B light chain 3B). 

## 2. Adipose Tissue and Cancer

Researchers have studied the link between white adipose and cancer for decades. Indeed, white adipose depots such as the omental fat pad are a common metastatic destination for many tumor cells [35]. Omental adipocytes have been shown to enhance invasiveness of gastric cancer cells by supplying oleic acid, which induced activation of PI3K/Akt signaling in the tumor cells; this process was associated with increases in the pro-invasion factor MMP−2 and was demonstrated both in vitro and using the chicken chorioallantoic membrane (CAM) model [36]. Omental adipocytes have also been shown to induce fatty acid binding protein 4 (FABP4) expression in ovarian cancer cells and promote metastasis and carboplatin drug resistance [37]. Knockdown of FABP4 resulted in increased 5-hydroxymethylcytosine levels in the DNA, downregulation of gene signatures associated with ovarian cancer metastasis, and reduced clonogenic cancer cell survival [37]. Interestingly, clustered regularly interspaced short palindromic repeat (CRISPR)-mediated knockout of FABP4 in high-grade serous ovarian cancer cells reduced metastatic tumor burden in mice [37]. The team also used a small-molecule inhibitor of the FABP family (BMS309403, which is thought to primarily target FABP4) and found that this not only significantly reduced tumor burden in a syngeneic, orthotopic mouse model, but also increased the sensitivity of cancer cells toward carboplatin both in vitro and in vivo, demonstrating the safety and efficacy of targeting FABP4 pharmacologically for ovarian cancer [37]. Mukherjee et al. also demonstrated that shRNAs against FABP4 reduced reactive oxygen species (ROS), beta-oxidation, lipid peroxidation, and proliferation, which were all increased in ovarian tumor cells when cultured with adipocytes [37]. Due to the complex effects of inhibiting FABP4 and the potential off-target effects of the pharmacological inhibitors, the exact mechanisms through which targeting the FABP family of proteins works are still unclear. FABP family inhibition may be decrease tumor progression through affecting fatty acid transport, metabolism, or localization, or due to the other metabolic or signaling effects.

Similarly, colon cancer also grows in an adipocyte-rich environment and locally advanced colon cancers often invade into surrounding adipose tissue, creating direct contact with adipocytes. A number of studies suggest that fatty acid transport and fatty acid oxidation (FAO) in the tumor cells are modulated by the surrounding adipose tissue and represent new target mechanisms. Indeed, in vitro co-culture experiments revealed increased FABP4 expression and lipid accumulation coupled with altered metabolism, invasion, and migration of colon cancer cells [38]. Pharmaceutical inhibition of FABP4 reversed the effects of the adipocyte co-culture, and conversely when FABP4 was overexpressed in colon cancer cells, significant increases in metastases were observed in vivo. A second study found that carnitine palmitoyltransferase I (CPT1A), an enzyme necessary in the beta-oxidation of long chain fatty acids, was upregulated in colon cancer cells exposed to adipocytes, while silencing of CPT1A eliminated the protective effect provided by adipocytes in vitro and in vivo [38]. Another recent study found that CPT1 contributes to antiangiogenic drug resistance, and that inhibition CPT1 significantly compromises free fatty acid (FFA)-induced cell proliferation using a variety of cancer types, including colorectral cancer and hepatocellular carcinoma cells [39]. Combined, these findings demonstrate the critical link between colon cancer cells, FAO, and adipocytes in the microenvironment.

In primary breast cancer, adipocytes and adipocyte-derived factors have been linked to drug resistance and malignancy. In vitro, experiments with HER2+ BT474 and SKBR3 breast cancer cells cultured in the presence or absence of conditioned media from mature adipocytes demonstrated that in basal conditions, tyrosine kinase inhibitors (TKI) (often used to treat HER2+ breast cancer) induced p27 leading to apoptosis, which did not occur in the presence of adipocyte conditioned medium [40]. Contact of breast cancer cells with adipose tissue in vivo also led to reduced sensitivity to TKI [40]. In acute lymphoblastic leukemia (ALL), ALL cells migrate toward mature adipocytes in vitro, as well as subcutaneous and visceral adipose explants, which provide protection from daunorubicin and vincristine [41]. In addition to their direct interactions with tumor cells in the microenvironment, adipocytes may also provide tumor cells with protection from chemotherapy through drug absorption and metabolism. In 2017, Sheng et al. demonstrated that adipocytes absorbed and metabolized daunorubicin, a treatment for ALL, reducing its active drug concentration, and thus its anti-tumor effect in the microenvironment [42]. Extracellular matrix (ECM) molecules of the omental fat pad, such as collagens, can also support tumor cell metastasis. For example, the expression of integrin alpha 2 (ITGA2) by ovarian cells contributes to their adhesion to collagen, and promotes cell migration, anoikis resistance, and peritoneal metastasis in vitro and in vivo [43]. ITGA2-dependent phosphorylation of focal adhesion kinase and activation of the mitogen-activated protein kinase pathway has been found to enhance oncogenic properties using phosphoproteomics [43]. It is likely that the ECM molecules of the BMAT depot similarly influence tumor cells in the bone marrow, but this is a largely under-studied area. In 2011, Dirat et al. characterized “cancer-associated adipocytes” (CAAs) in breast cancer, demonstrating that adipocytes can be hijacked by tumor cells, which causes them to exhibit a delipidated phenotype [44]. Cancer-associated adipocytes also had decreased expression of adipogenic genes, as well as overexpression of proteases and pro-tumor cytokines including IL−6, which the authors show is crucial in breast cancer invasiveness [44]. Since then, CAAs have been characterized in ovarian cancer [45], melanoma [46], pancreatic cancer [47], and recently myeloma [22,48], and are thought to provide fuel (by way of fatty acid substrates) and cytokine signals required for tumor growth and invasion. A full understanding of how CAAs affect overall disease progression or interact with other cells in the microenvironment (e.g., immune, stroma, or hematopoietic cells) is as yet widely uncharacterized.

Pre-adipocytes (adipose-derived mesenchymal stem cells (ADSCs) and bone marrow-derived MSCs) within adipose depots also appear to contribute to the supportive nature of adipose depots, although these progenitor cells seem to be even more supportive once they begin to interact with tumor cells and take on a “tumor-associated” phenotype, similar to cancer-associated fibroblasts (CAFs) or tumor-associated macrophages (TAMs) [49,50]. Indeed, ADSCs from the momentum of ovarian cancer patients, whether they had metastases or not, were found to express higher levels of α-smooth muscle actin (α-SMA) than ADSCs from patients with benign gynecologic disease [43]. Direct and indirect co-cultures showed that epithelial ovarian cancer cells induced ADSCs to express CAF markers, including α-SMA and fibroblast activation protein, via the transforming growth factor beta 1 (TGF-β1) signaling pathway [43]. Moreover, co-cultured ADSCs exhibited functional properties similar to those of CAFs, including the ability to promote ovarian cancer cells proliferation, progression, and metastasis both in vitro and in vivo [43]. Thus, when considering the role of BMAT in cancer progression, the importance of the pre-adipocytes and the effects of tumor cells on this population cannot be ignored.

## 3. BMAT and Multiple Myeloma

Aging and obesity are now well-known risk factors for multiple myeloma (MM), a blood cancer of the plasma cell, as well as its precursor disease, monoclonal gammopathy of undefined significance (MGUS) [51]. As obesity and aging also cause increases in BMAT, researchers have asked the logical question: does BMAT contribute to the risk of MM or contribute to the disease progression? A major question in the field is if and how aging, through effects on the BM niche, contributes to increased cancer risk (independent from increasing mutational burden within cells [52] and weakening the immune system [42]). This question is especially relevant in MM, where aging is a major risk factor and driver mutations are diverse and evolve chronologically and spatially [53].

The review “Signaling Interplay between Bone Marrow Adipose Tissue and Multiple Myeloma Cells” [54] provided an overview of the field of myeloma and BMAT in 2016. Since then, more evidence has demonstrated the many ways in which BMAds support myeloma cells. The Yang laboratory found that mature adipocytes in bone marrow protect myeloma cells against chemotherapy through autophagy activation [30] and that myeloma-associated adipocytes contribute to bone disease [48] through PPARγ, EZH2, and PRC2-related mechanisms. Moreover, they demonstrated that myeloma cells can shift the balance from osteoblastogenesis to adipogenesis through inhibiting the ubiquitin ligase MURF1 in MSCs [55]. This laboratory also found that the adipocyte-derived factor, resistin, induces multidrug resistance in MM cells by inhibiting cell death and upregulating the ABC transporter protein expression [56].

Leptin is a known pro-myeloma adipokine factor that is also increased in obesity. Analysis of leptin levels in humans identified significantly higher leptin in MM patients compared with normal controls, and found that leptin levels were positively correlated with MM clinical stage and other clinical predictors [31]. Not only does leptin support MM cell proliferation and reduced toxicity of bortezomib [31], but it also counteracts the anti-tumoral activity of invariant natural killer T (iNKT) cells, which express the leptin receptor [34]. Thus, leptin appears to both directly support tumor cells and indirectly support them through effects on the immune system, suggesting that targeting this obesity-related factor, either through decreasing obesity or blocking leptin–leptin receptor signaling, could be a promising therapy in MM or other obesity-related cancers.

Other adipokines, such as apelin [57] and chemerin [58], are suggested as diagnostic markers for MM, although the potential for targeting them therapeutically is unknown. IL−6 is one of the most well-known pro-myeloma adipokines, and targeting of IL6 signaling is under clinical investigation with monoclonal antibodies to *IL−6* (siltuximab) or the IL−6 receptor (tocilizumab) [59]. CCL2/monocyte chemotactic protein−1 (MCP−1) is an adipokine that promotes macrophage-associated chemoresistance in MM by shifting macrophages towards the M2-like phenotype [58]. Targeting nicotinamide phosphoribosyltransferase (visfatin) in tumor cells themselves using siRNA transfection has recently been shown to reduce cell proliferation and induce apoptosis, however the potential advantage of inhibiting visfatin derived from local or distant adipocytes is not known [39].

The role of fatty acid metabolism in MM has recently been reviewed [60], and researchers are currently exploring if targeting fatty acid metabolism in MM cells basally or when in co-culture with BMAT is a viable new therapeutic target. Evidence suggests that omega-3 fatty acids may have anti-myeloma effects via induction of apoptosis through both mitochondrial and cell death receptor pathways [61] or by reducing MM exosome-mediated suppression of natural killer (NK) cell cytotoxicity [62]. Similarly, arachidonic acid, a biologically active fatty acid, can induce apoptosis in chronic myeloid leukemia (CML) cells [63], and may induce ferroptosis-mediated cell death in MM [63]. However, other fatty acids may support myeloma cells through acting as inflammatory mediators or as a fuel source for MM cells, as has been seen in melanoma [64], and thus targeting fatty acid metabolism (for example with Triacsin C, an ACSL inhibitor) may represent a novel way to inhibit tumor cells [65]. In addition to targeting fatty acid metabolism, our laboratory has recently found that targeting fatty acid transport or uptake proteins may be another way to impede growth or drug resistance evolution in MM [26,66].

The Edwards laboratory recently built on their previous findings by demonstrating that myeloma cells downregulate the anti-myeloma protein adiponectin via TNFα [67]. This study also demonstrated an increase in BMAT in early disease or low tumor burden in mice [67], implicating a potential role for MM cells in accelerating adipogenic differentiation of progenitors. However, a recent study published by Mehdi et al. utilizing single-cell RNA-Seq suggests that myeloma-patient-derived MSCs are functionally quiescent and exhibit gene expression profiles consistent with impaired adipogenesis and enhanced angiogenesis compared to MSCs from normal donors [68]. These authors also identified a novel IGFBP2+-expressing population of small adipocytes that is suppressed in MM, which may contribute to reduced osteoblast differentiation and the vicious cycle of MM-induced bone disease. Our laboratory [22], as well as the Edwards laboratory [67], have found that BMAds shrink in the later stages of myeloma progression and that MM cells can have differing effects on adipogenesis based on tumor cell type or culture conditions [69]. We have also now demonstrated a senescence-associated secretory phenotype (SASP) in local adipocytes exposed to MM cells [22], adipocytes exposed before differentiation to MM cells [69], and in MSCs exposed to MM cells or from MM patients [70]. SASP proteins such as IL-6 can stimulate proliferation or drug resistance in myeloma cells, and thus targeting senescent bone marrow cells or senescence in general in the bone marrow niche represents a potential novel therapeutic target for MM patients. Table 2 summarizes ways in which myeloma cells and other tumor cells can hijack and modulate BMAds.

## 4. BMAT and Breast and Prostate Cancer

A recent review has discussed the roles of BMAds (and leptin, adiponectin, and Sam68 specifically) in bone metastasis from breast cancer [19]. There are conflicting results when analyzing the influence of age in the development of bone metastasis for breast cancer patients [74]. Obesity, however, is a strong risk factor for developing breast cancer, and although one study found that obesity had no effect on the recurrence pattern of early breast cancer patients, it did find that obese early breast cancer patients had shorter overall survival compared to their normal-weight counterparts [75]. To analyze the role of BMAds in bone metastasis specifically, the King laboratory has performed elegant work demonstrating that breast cancer cells migrate to BMAds and interact with them closely using human tissue femur explants [76,77]. Using co-injections of breast cancer cells and adipocytes or pre-adipocytes into nude mice, BMP9 was found to inhibit the growth and metastasis of breast cancer cells, which may be in part related to their interaction with pre-adipocytes or adipocytes via leptin signaling [71].

The Podgorski laboratory demonstrated that prostate and breast cancer cells exposed to adipocyte-rich environments increase HO−1, an oxidative stress enzyme, which contributes to tumor growth and invasiveness. Adipocytes also induced expression of the endoplasmic reticulum (ER) chaperone BIP and splicing of XBP1, indicating adipocyte-driven unfolded protein response, which was sensitive to antioxidant treatment. Survivin expression in tumor cells was found to contribute to tumor cell survival in response to oxidative and ER stress or HO-1 induction by adipocyte exposure [33]. This laboratory also demonstrated the importance of using 3D culture systems wherever possible to co-culture adipocytes and prostate cancer cells, similar to what we have seen in BMAT and myeloma 3D cultures [13], to create a more physiologically relevant culture system [78]. Recently, the Podgorski laboratory demonstrated that prostate tumor cell-derived IL1β can induce an inflammatory phenotype in BMAds and decrease sensitivity to docetaxel via lipolysis-dependent mechanisms [78]. This team also observed that CXCL1 and CXCL2 derived from BMAds contribute to osteolysis in metastatic prostate cancer [79]. These data suggest that BMAds contribute to cancer-associated bone disease, which is a very interesting avenue to pursue.

The Podgorski laboratory also found that BMAds promote a Warburg phenotype (i.e., extensive glycolysis even in the presence of oxygen) in metastatic prostate tumors via HIF-1α activation and demonstrated that the lipolytic enzymes Adipose triglyceride lipase (ATGL) and hormone-sensitive lipase (HSL) are upregulated in BMAds exposed to prostate cancer cells, which paralleled a release of FFAs from the BMAds [32]. Although all FFAs can act as a fuel source for FAO, the nature and properties of the FFA released from BMAds may be very important, as inflammatory fatty acids may specifically increase prostate cancer bone metastasis. For example, Wang et al. found that the total level of FFAs and caprylic acid (C8:0) were significantly higher in prostate cancer patients with bone metastases, and demonstrated in vitro with co-culture systems that caprylic-acid-treated adipocytes promoted the invasion and migration of prostate cancer cells [78]. As described with MM above, alterations in lipid metabolism are also observed in breast tumors at both the cellular and tissue levels [80]. Targeting lipid metabolism in cancer cells in the bone marrow holds great promise, but more analyses (specifically metabolic flux, proteomic or lipidomic mass spectrometry, gene expression, and functional outputs) are needed in tumor cells treated with different FFAs before safely and effectively targeting fatty acid metabolism in preclinical and clinical settings can become a reality.

## 5. BMAT and Leukemia

Data are accumulating showing that leukemic cells are also protected from chemotherapy-induced cytotoxicity when in close proximity to BMAds through an array of mechanisms. This topic has been specifically covered in a comprehensive review from Dr. Fischer-Posovszky’s group [42], in which they raise the question about the role of BMAds with the title “Adipocytes in Hematopoiesis and Scute Leukemia: Friends, Enemies, or Innocent Bystanders?”. It appears that BMAds have temporally and disease-state-dependent roles in acute myeloid leukemia (AML) progression and myelo-erythroid maturation. Adipocytes can sequester and metabolize the chemotherapy daunorubicin [42], and this is in part due to a complex loop whereby adipocytes protect acute lymphoblastic leukemia (ALL) cells from oxidative-stress-induced cell death (which can also be mimicked by treating the cells with antioxidants), while ALL cells induce oxidative stress in adipocytes, which leads to their secretion of pro-survival factors that protect tumor cells from daunorubicin [42]. The team also found that glutathione synthesis is partially the cause of BMAd protection of ALL cells [42].

However, leukemic suppression of BMAds has also been shown by Boyd et al. to lead to imbalanced regulation of endogenous hematopoietic stem and progenitor cells, resulting in impaired myelo-erythroid maturation [72]. This team found that in vivo administration of PPARγ agonists induced BM adipogenesis, which rescued healthy hematopoietic maturation while repressing leukemic growth. The study was the first to identify an unappreciated axis between BM adipogenesis and normal myelo-erythroid maturation that could potentially have therapeutic implications for BM failure in AML by non-cell autonomous targeting of the niche [72].

Similarly to that described in later stage MM, AML patients have been found to have fewer and smaller adipocytes in their BM sections, as based on perilipin immunohistochemistry (IHC), compared to controls [81]. An adverse effect of smaller adipocytes on AML patient prognosis has been observed [82]. Similarly to that described in other cancers, FABP4 appears to be a potential target in AML, as it promotes AML aggressiveness through enhanced DNMT-1-dependent DNA methylation [83,84]. The Shi group also found that leukemic cells can produce growth differentiation factor 15 (GDF15), which remodels the residual BMAds into small adipocytes and is associated with a poor prognosis in AML patients, and that transforming growth factor-β type II receptor (TGFβRII) is the main receptor for GDF15 on BMAds [71]. They showed that transient receptor potential vanilloid (TRPV) channels negatively regulated GDF15-induced remodeling of BMAds, and found that GDF15 reduced the expression of the transcription factor Forkhead box C1 (FOXC1) in BMAds [71]. Shafat et al. demonstrated that AML blasts program BMAds to generate a protumoral microenvironment as well [85]. AML cells induced adipocyte lipolysis of triglycerides via increased HSL activity, which released FFAs. Upregulated FABP4 in both AML cells and adipocytes was observed when they were in co-culture, which increased tumor cell proliferation, as validated by short hairpin RNA knockdown experiments [85]. They also found that inhibiting CPT1A in an AML-patient-derived xenograft model improved mouse survival [85], again supporting that fatty acid metabolism is a pro-survival metabolic pathway for tumor cells in the bone marrow. Also in AML, Tabe et al. demonstrate that tumor cells are protected from spontaneous apoptosis via upregulation of FAO and from increased AMP-activated protein kinase (AMPK) signaling resulting from FFA transfer from BMAds [28]. The recent review by this group is an excellent resource on this topic and highlights the recent progress in our understanding of fatty acid metabolism in AML cells in the adipocyte-rich BM microenvironment [71].

## 6. Conclusions and Discussion

It is clear that BMAds can contribute to cancer through a variety of mechanisms, which fuels the quest in the field to understand and target BMAds or factors derived from these cells. Bone marrow adipogenesis and mature BMAds are inextricably linked to osteogenesis and bone turnover (i.e., osteoclast activity, immune or hematopoietic cell activity, angiogenesis or vasulogenesis, neuronal signaling, and systemic endocrine system function). The Morrison laboratory has shown that A-ZIP/F1 “fatless” mice have delayed hematopoietic regeneration due to a decrease in stem cell factor (SCF), which is necessary for vascularization of the marrow before hematopoiesis [24]. Recent work from Teitelbaum and colleagues has demonstrated that the ablation of mature fat cells using an adiponectin-Cre–diphtheria toxin receptor (DTR) mouse model induces extensive bone acquisition [86]. Demonstrating the bidirectional communication between these cells, factors that often modify bone directly or indirectly modify BMAT [87,88]. For example, we have demonstrated that removal of bone cells can increase BMAT content in a mouse osteocalcin-Cre-inducible DTR model, depending on the gender and extent of bone cell removal [89].

There are many connections between bone and BMAT, for example through the Wnt pathway. The bone-building sclerostin-neutralizing antibody changes MSC fate by increasing Wnt signaling, decreasing adipogenesis, and increasing osteogenesis preferences [90]. The sclerostin-neutralizing antibody decreases BMAT in vivo in several conditions, likely through indirect and direct effects, as also examined in vitro [87]. Interestingly, increased bone volumes and decreased BMAT using the sclerostin-neutralizing antibody may not only improve bone disease, as has been seen in myeloma [91,92], it may also potentially decrease metastasis for certain tumors, such as breast cancer [93]. Bone cells also produce the protein osteonectin, or SPARC (secreted protein acidic and rich in cysteine), an extracellular matrix glycoprotein which was recently shown to inhibit adipocyte-induced homing, proliferation, and invasion of ovarian cancer cells [94]. SPARC also suppressed metabolic programming of ovarian cancer cells and inhibited adipocyte differentiation and their phenotypic switch to a cancer-associated phenotype [94]. Mechanistic studies revealed that this effect was mediated through inhibition of cEBPβ–NFkB–AP1 transcriptional machinery [94]. Thus, switching the fate of MSCs from adipocyte to osteoblastic lineages may be a promising research avenue for cancers in the bone.

One conundrum in the field is that tumor cells themselves can decrease BMAT [22,67], suggesting that removing the BMAT may not be the answer to how best to “target” BMAT and that a more nuanced approach (e.g., removing certain types of BMAds or targeting more specific BMAT secreted proteins, fatty acids, or cell contact proteins) may be a more effective way of targeting the BMAT niche in cancer. Novel software and imaging tools, such as MarrowQuant [19] or ImageJ pipelines [95], will help to better understand BMAd biology and response to tumor cells to help understand how BMAT responds to tumor cells.

One important outstanding question is how lipid droplets, which may accumulate (or be utilized and decrease) in tumor cells when co-cultured with adipocytes, contribute to disease progression. Lipid droplets play roles in immunosuppression, inflammation, angiogenesis or vascular permeability, lymphangiogenesus, organotropism, and cellular reprogramming [36]. Our laboratory has observed that either increased or decreased lipid droplet accumulation can occur in tumor cells, highlighting an avenue that requires further understanding. Overall, targeting BMAT and its products, such as fatty acids, growth factors, cytokines, chemokines, matrix molecules, and metabolites, is an important avenue for basic and translational research. The field will require a broad range of chemists, biologists, bioinformaticians, clinicians, and engineers to understand how to target this depot and develop novel ways to carry this idea into the clinic.

## Figures and Tables

**Table 1 cancers-13-00364-t001:** Factors and mechanisms through which tumor cells are affected by BMAds.

Factor	Example	References
Cytokines/Chemokines/Adipokines	IL−6, IL−17, MCP1	[22,23]
Growth Factors	SDF−1 (Cxcl12), SCF (Kitl), IL−7 (Il7), IL−15 (Il15), IL−34 (Il−34), M-CSF (Csf1), BMP−4 (Bmp4), and CCL−19 (Ccl19)	[24,25]
Fatty Acid Binding Proteins	CD36, FABP4, FABP5	[26,27,28]
Lipid Mediators	Free Fatty Acids	[28]
Fatty Acid Metabolism Enzymes	CPT1	[28]
RNAs	microRNAs (potentially)	[29]
Metabolites or Hormones	Leptin, Adipsin	[30,31]
Warburg Effect	via HIF1a	[32]
Oxidative and ER Stress Pathways	Regulation of Heme Oxygenase 1 and Survivin	[33]
Autophagy activation	Jak/Stat3 Induction of Atg3, Atg5, and LC3-I/II	[30]
Adipokines Suppress Immune System Function	Leptin Reduces Anti-MM Effects of Invariant Natural Killer T (iNKT) cells	[34]

**Table 2 cancers-13-00364-t002:** Factors and mechanisms through which BMAds are affected by tumor cells.

Factors	Example	References
Induction of Lipolysis (Shrinking of Adipocytes)	Adipose triglyceride lipase (ATGL) and phosphorylation and activation of hormone-sensitive lipase (HSL)	[32,71,72]
Induction of Senescence-Associated Secretory Phenotype	IL6, CXCL1, CXCL2	[22]
Pro-Tumor Factors	COX−2 and MCP−1	[73]
Decreased Anti-Tumor Hormones	Adiponectin	[67]
Mature Adipocyte Size Can Increase in Early Stages of Disease. MSCs Skewed from Osteogenesis to Adipogenesis.	α4 integrin subunit on MM cells stimulating VCAM1 On MSCs causing activation of Pkcβ1 signaling and repression of the muscle ring-finger protein−1 (MURF1)-mediated ubiquitylation of Pparγ2	[23,55,67]

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
