# Peer review of "Bone Marrow Adipocytes: A Link between Obesity and Bone Cancer"

_cancers, 2021, doi:10.3390/cancers13030364_

Round 1

Reviewer 1 Report

This is a very nicely written review, summarising recent advances in the field of bone marrow adiposity and cancer. My only real comment is that there is not much discussion on ageing and cancer, the review largely reads like a review on BMAT and cancer, with a few mentions of the links with aging and cancer. Personally, I would either expand the aging or tone down the title.

Table 1: Tumour cells can be affected by adipokines from BMAds, this should be included in this table

Author Response

This is a very nicely written review, summarising recent advances in the field of bone marrow adiposity and cancer. My only real comment is that there is not much discussion on ageing and cancer, the review largely reads like a review on BMAT and cancer, with a few mentions of the links with aging and cancer. Personally, I would either expand the aging or tone down the title.

Table 1: Tumour cells can be affected by adipokines from BMAds, this should be included in this table

Thank you for these suggestions. We agree with the reviewer that the word Aging should be removed from the title, and we have done that. We also have added the word Adipokines to Table 1 and added more discussion of the roles of certain adipokines to the main text, as highlighted with red text. We appreciate these comments!

Reviewer 2 Report

The authors raised an interesting topic of the association of adipocyte. adipocyte tissue and cancers. Though the issue has novelty, there are some issues for the publication.

  1. The manuscript is not enough for a full review. There must have more data and content
  2. Though the authors reviewed this interesting topic, there are no thoughtful theory for the association between the adipocytes tissue and the cancers.
  3. The title state about the aging, however, little has been mention about aging in the text.

Author Response

The authors raised an interesting topic of the association of adipocyte. adipocyte tissue and cancers. Though the issue has novelty, there are some issues for the publication.

We thank this reviewer for the excellent suggestions and important criticisms of our prior manuscript. We have made edits in the paper accordingly and most edits are highlighted in red text in the manuscript.

1. The manuscript is not enough for a full review. There must have more data and content

Q1) We appreciate this direct and honest feedback from Reviewer 2. We have taken this feedback and added more content including more data (specifically about different adipokines and how they affect tumor cells), references, and discussion. We have added this a lot to the introduction section about differences between BMAds and other adipocytes as well.  

2. Though the authors reviewed this interesting topic, there are no thoughtful theory for the association between the adipocytes tissue and the cancers.

Q2) We agree that the review would be improved with more thoughtful analysis of the theory about the interactions between adipocytes and cancers, based on our analysis of the literature. We have now added more of that throughout the manuscript. We have also discussed the effects that obesity/increased BMAT can have on the immune system, and how that could translate to effects on cancer cells, which at this point in the field is more speculative, but which is important to consider and investigate. We also suggest to the reader that the importance of the adipocyte used in these studies and describe new findings about differences between BMAT and WAT, which are not inconsequential.

3. The title state about the aging, however, little has been mention about aging in the text.

Q3) Lastly, we agree that the title was misleading and removed the word Aging from the title.

Reviewer 3 Report

This review article highlights a blossoming field of BMAT in supporting tumor growth and invasion. The review article is timely, well-written, and informative by discussing the latest publications in the field. 

The authors should consider to include the following aspects in the revised version of manuscript.

1) What are the real differences between BMAds and Ads either BAT or WAT located elsewhere ? Metabolic differences, endocrine differences, and signaling differences should be discussed to allow authors better understanding this specific depot. Ideally, the authors should prepare a table to list the known differences between BMAds and other Ads.

2) While this reviewer appreciates the authors´effort and knowledge in describing various findings from different laboratories, most known knowledge about Ads in promoting tumor growth and metastasis are from non-BMAds. The listed factors also exist in other adipose tissues. What are the specific roles of BMAds in cancer progression? Do they promotes local tumor growth, metastasis, and drug responses? The authors are encouraged to speculate their visionary thoughts.  For example, they may explain why bone cancers are so difficult to treat. If these tumors grow in another organs or tissues would they become easier to treat.  Has anybody done this type of experiment?

3) Some literature should be cited, including

  • PMID: 29861385

4) Abstract: "life expenctacies" should be "life expectancies". There are some other places need to be fixed.  

After minor revision, the well-written manuscript should be accepted. I should congratulate the authors on writing this wonderful piece.  

Author Response

This review article highlights a blossoming field of BMAT in supporting tumor growth and invasion. The review article is timely, well-written, and informative by discussing the latest publications in the field. 

The authors should consider to include the following aspects in the revised version of manuscript.

We thank this reviewer for the excellent suggestions. We have made edits in the paper accordingly and most edits are highlighted in red text in the manuscript.

1) What are the real differences between BMAds and Ads either BAT or WAT located elsewhere ? Metabolic differences, endocrine differences, and signaling differences should be discussed to allow authors better understanding this specific depot. Ideally, the authors should prepare a table to list the known differences between BMAds and other Ads.

Q1) We thank the reviewer for the question on the differences between BMAds and other adipocytes. We have now included a section on this, which is also copied here below. We have also included references to the paper that has the data available in a Supp. Table of all 897 genes differentially expressed between WAT and BMAT by microarray gene expression from a 2018 Scientific Reports manuscript by Craft et al.

It is important to remember that bone marrow adipocytes are distinct from other white or brown/beige adipocytes, as can be seen in its distinctive response to starvation and β3-agonist stimulation, and its unique roles in glucose homeostatis [3,45,46]. The biology of BMAds is further complicated by the fact that different bone marrow locations (eg. vertebrae, femur, and tibia; distal versus proximal) have different types of BMAds, which respond differently to a variety of stimuli [1,47]. However, BMAds are also similar to white adipocytes in that they can store and release fatty acids and adipokines (such as adiponectin), and they are predominantely unilocular [46]. Hundreds of genes and dozens of pathways have been found to be differentially expressed or active in BMAT vs WAT, based on microarray gene expression analysis of rabbit tissues [46]. In this analysis, Craft et al. found that BMAT metabolism showed differences from WAT metabolism, including decreased glycerol content, insulin resistance, decreased fatty acid metabolism, and decreased thermoregulation. Genes related to lipid synthesis and transport pathways (such as leptin, FABP4, FABP5, and interleukin 6 signal transducer were also reduced in BMAT, although still expressed, while cellular pathway analysis suggested reductions in fatty acid β-oxidation and oxidative phosphorylation [46]. Other genes were found to be increased in BMAT vs WAT, such as caveolin family proteins, collagens, and integrins, among many other genes, and the Inguinity Pathway Analysis identified PPAR signaling as increased in BMAT [46,48].

2) While this reviewer appreciates the authors´effort and knowledge in describing various findings from different laboratories, most known knowledge about Ads in promoting tumor growth and metastasis are from non-BMAds. The listed factors also exist in other adipose tissues. What are the specific roles of BMAds in cancer progression? Do they promotes local tumor growth, metastasis, and drug responses? The authors are encouraged to speculate their visionary thoughts.  For example, they may explain why bone cancers are so difficult to treat. If these tumors grow in another organs or tissues would they become easier to treat.  Has anybody done this type of experiment?

Q2) We also thank the reviewer for asking the question about how tumor cell support from BMAds compares to support from other types of adipocytes (visceral or subcutaneous). We have added discussion about this. We agree with the reviewers that many of the studies that exist have used other types of white adipocytes, but not necessarily bone marrow adipocytes specifically. We have mentioned that a study comparing different effects of different adipocytes would be greatly beneficial to the field. We have added also more data about the effects of adipocyte/adipokines on tumor cells that may explain the supportive nature of adipocytes on tumor cells. Other new content has also been added to strengthen our manuscript.

3) Some literature should be cited, including

  • PMID: 29861385

Q3) Thank you for the suggestion to add the following paper: PMID: 29861385, Iwamoto et al. Cancer Lipid Metabolism Confers Antiangiogenic Drug Resistance. We’ve now added that.

4) Abstract: "life expenctacies" should be "life expectancies". There are some other places need to be fixed.  

Q4) Thank you for letting us know about the issues in the abstract. That has been corrected and other edits of the paper have been made in grammar/spelling/typos.